# Biodiesel Production Using Palm Oil with a MOF-Lipase B Biocatalyst from Candida Antarctica: A Kinetic and Thermodynamic Study

**DOI:** 10.3390/ijms241310741

**Published:** 2023-06-28

**Authors:** Liliana Giraldo, Fernando Gómez-Granados, Juan Carlos Moreno-Piraján

**Affiliations:** 1Facultad de Ciencias, Departamento de Química, Universidad Nacional de Colombia, Bogotá 111231, Colombia; lgiraldogu@unal.edu.co (L.G.); fagomezg@unal.edu.co (F.G.-G.); 2Grupo de Investigación en Sólidos Porosos y Calorimetría, Facultad de Ciencias, Departamento de Química, Universidad de los Andes, Bogotá 111711, Colombia

**Keywords:** Candida Antarctica, biodiesel, kinetics, lipase, adsorption, thermodynamic, metal-organic frameworks, isotherms

## Abstract

This research presents the results of the immobilization of Candida Antarctica Lipase B (CALB) on MOF-199 and ZIF-8 and its use in the production of biodiesel through the transesterification reaction using African Palm Oil (APO). The results show that the highest adsorption capacity, the 26.9 mg·g^−1^ Lipase, was achieved using ZIF-8 at 45 °C and an initial protein concentration of 1.20 mg·mL^−1^. The results obtained for the adsorption equilibrium studies allow us to infer that CALB was physically adsorbed on ZIF-8 while chemically adsorbed with MOF-199. It was determined that the adsorption between Lipase and the MOFs under study better fit the Sips isotherm model. The results of the kinetic studies show that adsorption kinetics follow the Elovich model for the two synthesized biocatalysts. This research shows that under the experimental conditions in which the studies were carried out, the adsorption processes are a function of the intraparticle and film diffusion models. According to the results, the prepared biocatalysts showed a high efficiency in the transesterification reaction to produce biodiesel, with methanol as a co-solvent medium. In this work, the catalytic studies for the imidazolate, ZIF-8, presented more catalytic activity when used with CALB. This system presented 95% biodiesel conversion, while the biocatalyst formed by MOF-199 and CALB generated a catalytic conversion percentage of 90%. Although both percentages are high, it should be noted that CALB-MOF-199 presented better reusability, which is due to chemical interactions.

## 1. Introduction

From the industrial revolution to today, humanity has based all its developments on energy consumption, using fuels that pollute the environment. Therefore, it is up to science and engineering to generate solutions to find alternative, viable, clean, and scalable energy sources. Among these alternatives, exciting and essential biofuels [1,2,3,4,5] that are economical and environmentally friendly have been investigated. The use of biofuels was proposed several decades ago as a solution to these drawbacks, but only recently has more attention been paid to this alternative. Ethanol became important not only because of its economic advantages but also because the Kyoto Protocol changed the dynamics of oil prices. The main objective of biofuel preparation processes is to obtain a new energy source through unique processes that generate a positive economic, social, and environmental impact [5,6]. Biofuels are substances produced from biomass and serve as a renewable energy source, reducing CO_2_ production [6,7,8]. To reduce the cost of biodiesel production by methanolysis [1,2,3,4], the process of biodiesel production by transesterification of oils catalyzed by lipases has begun to be used, a method that has been very interesting as unrefined oils can be used, generating fascinating results.

It should be noted that using lipases allows for mild operating conditions, low energy requirements, and easy product separation [3,4]. In this research, Candida Antarctica Lipase B (CALB) was selected as it is much more thermostable than most lipases and does not depend on metal cations. It also has adequate activity [3,4,5]. Much remains to be studied in this regard, and many thermodynamic aspects need to be clarified during biodiesel synthesis when using this type of association with lipases. [4,5]. It is characterized by high activity in the presence of a wide range of non-natural esters and good movement against large triglycerides. In addition, it is a highly specific lipase, especially in hydrolysis processes [4,5] such as the one that will be used in this work [4,5].

Starting with X-ray diffraction spectroscopy studies to elucidate the crystal structure of Candida Antarctica Lipase B (CALB), Professor J. Uppenberg [8,9], together with his laboratory collaborators, demonstrated that the enzyme has a catalytic triad Ser-His-Asp in its active site. They also suggested that the structure possibly appears to be in an “open” conformation with a relatively restricted entrance to the active site, which in their research explains the substrate’s high specificity and the high degree of stereospecificity in this CALB lipase, which is classified as a globular-type protein, specific to a comprehensive class of esters, and belongs to the α/β hydrolase family [5,6].

Several colleagues have published interesting studies in specialized laboratories where metal-organic networks (MOF) such as ZIF-8, MOF-5, MOF-253, and IRMOF-16 are used to immobilize different lipases using the level adsorption method to synthesize biodiesel [10,11,12,13,14,15,16,17,18]. In the literature, values have been published regarding the yield of biodiesel production between 80 and 85% after 18 h of reaction in an alcoholic medium, in several steps at a constant temperature, verifying that the procedure of fixing the Lipase is a function of the variables, temperature, pH, and topography of the MOF used. An alternative method to link the support and the Lipase has also been reported called chemisorption. In this method, the structure of the chemical and structural composition of the Lipase are joined by covalent bonds to the MOFs by explicitly using the molecules that surround the central metal of the structure [19]. This allows stable biocatalysts to be obtained, and the Lipase is prevented from migrating into the solution, making the structure slightly more rigid [20,21,22,23]. The fact that the enzyme is immobilized through covalent bonds makes it more efficient, although it should be noted that the physically linked one is usually more active [17]. The literature has reported that the binding of enzymes to the internal pores of MOFs may not be efficient. This is because the lipase migrates (leaches), which is a phenomenon associated with the physical adsorption process. This directly affects chemical adsorption, which usually prevents lipase activity from decreasing, which may negatively affect enzymatic activity. Mesoporous MOFs, such as MIL-100(Fe) and MOF-199, have been used to serve as anchors in the immobilization process through processes such as coprecipitation [20,21]. When this type of procedure is carried out this way, the Lipase will remain inside the porous network of the MOFs during the synthesis process. Studies of lipase encapsulated with MOF that have been successfully used in the transesterification reaction have shown excellent thermal stability compared to free Lipase [14,22,23]. Some studies have reported the encapsulation of lipases within MOFs with the primary objective of performing biodiesel synthesis during transesterification and using lipase encapsulation to obtain good yields, including various cycles of use [24,25,26].

Thermodynamic and kinetic studies on the pre-supported lipase-MOF system are scarce, and it is essential to have this data available [17]. The analysis presented in this investigation closely follows the experimental part (making some changes to adjust to our conditions) for comparison purposes [27,28].

Therefore, the scope of this research is not only to synthesize two Lipase-MOF biocatalysts and to characterize them texturally and chemically, but also to study the mechanisms of lipase adsorption (*Candida Antarctica B Lipase*) on the MOFs: ZIF-8 and MOF-199. Additionally, using models of two and three variables, this work demonstrates how the Lipase is adsorbed on the MOF’s.

Finally, this work contributes to kinetic and thermodynamic studies concerning the use of biocatalysts for biodiesel synthesis.

## 2. Results and Discussion

### 2.1. MOFs Characterization

#### 2.1.1. Analysis of FTIR Results

The corresponding FTIR results for the free MOF-199 and CALB-MOF-199 samples are shown in Figure 1a. The corresponding FTIR results taken for samples MOF-199 free and CALB-MOF-199 are shown in Figure 1a, while Figure 1b presents the FTIR results for ZIF-8 and CALB-ZIF8.

The spectra of the MOFs with the Lipase were taken at 45 °C with a CALB load of 0.40 mg·mL^−1^. The analysis shows that each one of the bands of the respective spectra starting with free MOF-199 (Figure 1a—line black) generally coincides with what has been reported by various authors in the specialized literature [28,29]. The following characteristic bands stand out from the spectrum: 1285 cm^−1^, which could be due to CO vibrations of BTC (benzene-1,3,5-tricarboxylate), and the bands at 1456 cm^−1^ and 1340 cm^−1^, which could be due to the vibration and in-plane splitting of the C=O of BTC (benzene-1,3,5-tricarboxylate). Suppose one compares the FTIR of free MOF-199 with CALB-MOF-199. In that case, it is observed that there is a very marked band for the latter system, which some authors call “characteristic” at 1355 cm^−1^ and assign it to the aromatic C-C stretching vibration of the BTC group of the carboxylate group.

In addition, a comparison of the FTIR of free MOF-199 with CALB-MOF-199, a band that can be assigned to the -OH group, is presented to be above 3000 cm^−1^, at which point the intensity becomes more intense for CALB-MOF-199, clearly demonstrating the effect of lipase immobilization on the adsorbate. This allows us to confirm that the procedure carried out in the laboratory regarding the fixation of the Lipase on the MOF-199 was successful. The FTIRs presented in Figure 1b show the spectra of ZIF-8 free with Lipase adsorbed at 45 °C and 0.40 mg·mL^−1^ of lipase loading. (CALB-ZIF-8). The results of the analysis were found to be in good agreement with those from previous studies [28,29]. The bands that lie towards approximately 440 cm^−1^ can be assigned to vibrations of the Zn-N pair. Suppose the infrared spectrum is analyzed from left to right. In that case, certain acute vibrations will be found, in particular two bands of 770 cm^−1^ and 1500 cm^−1^, which experts assign to the HMeIM ring, and some very small bands towards the weak 2800–3100 cm^−1^ that surely correspond to the stretching and vibrations of aromatic and aliphatic CH of HmeIM [18,28]. The results of the analysis based on the FTIR presentation of the band show that they correspond to ZIF-8 in its hexahedral crystalline form together with adsorbed Lipase, as shown in the plots in Figure 1b.

#### 2.1.2. Analysis of X-ray Diffraction Analysis (XRD) Results

This section presents the results obtained using the XRD technique to characterize the crystalline structures of the two MOFs used in this research, both free and with immobilized Lipase. In Figure 2a the XRD results for free MOF-199 and the CALB-MOF-199 system, respectively, are presented. The XRD spectra in Figure 2a for MOF-199 present very well-defined diffraction peaks that agree with what has been reported by some authors who have simulated this X-ray diffraction spectrum, as well as with what has been reported in the specialized literature [28,30,31,32], which shows that the MOF-199 used in this work corresponds to the crystalline structure characteristic of these metal-organic lattices. Outstanding and very well-defined peaks can be highlighted by “sweeping” the spectrum from right to left: 6.5°, 9.3°, 11.7°, 13.7°, 19.0° and 25.8° are assigned to the crystalline planes whose Miller’s indices are: (2 0 0), (2 2 0), (2 2 2), (2 2 1), (4 0 0), (4 4 0), and (7 3 1), respectively. When the adsorption procedure fixes CALB on MOF-199 (see the same Figure 2a, black color), the positions of the main diffraction peaks remain consistent with those of MOF-199 (i.e., free, as presented in the same Figure 2a in grey color).

The XRD of MOF-199 free and with Lipase shows a slight reduction in the peaks. This decrease in the size of the peaks when the Lipase is added may be due to several factors. The most published reason for this, argued by several authors, is that there is little space between the atomic layers in the crystalline material due to the presence of the enzymatic molecules on the surface [18,28], which gives rise to a decrease in the crystallinity of MOF-199 after CALB immobilization.

On the other hand, the results of the XRD patterns for free ZIF-8 and ZIF-8 with adsorbed Lipase (CALB-ZIF-8) are shown in Figure 2b—black—and Figure 2b—grey, respectively. Figure 2b shows the XRD of free ZIF-8, which agrees well with values reported in the scientific literature [28,33]. The XRD for this sample presented the following most significant peaks when the spectrum was scanned from left to right in the diffractogram: the main diffraction peaks mean the intensification for the ZIF-8 position at 2θ of 7.28°, 10.8°, 13.0°, 15.0°, 16.8°, and 18.8°, which, when calculating the Miller indices, correspond to the crystalline planes (0 1 1), (0 0 2), (1 1 2), (0 2 2), (0 1 3), and (2 2 2), respectively [28,34]. By examining and analyzing the XRD diffractograms corresponding to the CALB-ZIF-8 samples, shown in Figure 2b—grey, the XRD of the sample, as in the XRD of MOF-199 (already analyzed), the immobilization of Lipase does not affect the crystallinity of ZIF-8. Notably, there is only a slight change in the XRD spectrum after lipase immobilization, indicating that Lipase was successfully immobilized on ZIF-8. The main diffraction peaks signifying the intensification for the 2θ position of ZIF-8 are: 28°, 10.8°, 13.0°, 15.0°, 16.8°, and 18.8°, which, when calculating the Miller indices, correspond to the crystalline planes (0 1 1), (0 0 2), (1 1 2), (0 1 2), (0 2 2), (0 1 3) and (2 2 2), respectively [28,34]. 

#### 2.1.3. Analysis of the Porosity and Surface Area Results

Once the respective isotherms of N2 at 77 K have been run for the two adsorbents MOF-199 and ZIF-8 and the isotherms of the biocatalysts CALB-MOF-199 and CALB-ZIF-8. These are shown in Figure 3, thus:

Figure 3a, black circles and open circles represent the adsorption-desorption isotherm of MOF-199; In this same graph you can see the isotherm for CALB-MOF-199, which is shown with gray circles (adsorption) and open circles (desorption). 

Figure 3b, Gray circles corresponding to adsorption-desorption, where in this case there is no hysteresis loop for the ZIF-8 sample and the black squares show adsorption-desorption (a slight hysteresis loop is observed) for the ZIF-8 sample. CALB-ZIF-8 sample.

A summary of the results is shown in Table 1. It can be seen that the BET-specific surface areas for the MOF-199 and ZIF-8 were 1750 m^2^·g^−1^, 1733 m^2^·g^−1^, respectively.

The results summarized in Table 1 show that ZIF-8 has a smaller BET surface than MOF-199, but it has a larger pore radius, a larger micropore volume, and a larger mean pore size. On the other hand, it is worth noting that the MOFs that were used to support the CALB lipase present Type I isotherms (see Figure 3a,b) that, according to recent IUPAC recommendations, classify the isotherms [35], which correspond fundamentally to solids of a microporous nature (size < 2 nm).

The micropore volume is 0.88 cm^3^·g^−1^, calculated by applying the Dubinin–Astakhov (DA) method for ZIF-8, while for MOF-199, it is 0.66 cm^3^·g^−1^. As mentioned, ZIF-8 is a microporous material with a BET-specific surface area of 1733 m^2^·g^−1^ (relative pressure calculated from 0.009 to 0.02), while that of MOF-199 is 1750 m^2^·g^−1^.

These results indicate that the two MOFs have textural properties that allow them to receive adsorbates in their structure, due to micropores and the incipient presence of mesoporosity that develops in ZIF-8, and to their highly-specific surface area. On the other hand, as shown in Figure 3a—grey, the N_2_ isotherm at 77 K corresponds to the CALB-MOF-199 biocatalyst. This isotherm also has a type I isotherm characteristic corresponding to fundamentally microporous systems with a SBET that has decreased and is now 1400 m^2^·g^−1^, as well as a pore radius that is now 5.5 Å; when analyzing the results in Table 1, where the other textural parameters are reported, it is clear that these change when CALB is configured in MOF-199, and the values of each of its parameters decrease. It is worth noting here that the CALB-MOF-199 biocatalyst also exhibits a type I isotherm, which suggests that the basic framework of MOF-199 is conserved after CALB immobilization. Figure 3b—black—shows the isotherm obtained for the CALB-ZIF-8 biocatalyst. It behaves very similar to the free ZIF-8 isotherm (Figure 3a–grey), i.e., it is microporous with a type I isotherm according to IUPAC. In Table 1, it is again concluded, from the point of view of its textural characteristics, the parameters of this biocatalyst change when CALB is supported; the S_BET_ now has a value of S_BET_ 1350 m^2^·g^−1^ and the pore radius is 7.2 Å. The other textural parameters decrease due to the adsorption of CALB, as in the previous biocatalyst. To summarize the above information, it has been established that the reduction in the BET surface area when fixing the CALB on the two MOF’s may be due to the impregnation and/or blockage of the pores by the Lipase inside the MOF’s pores. This agrees with the reduction in the value of the parameters calculated through the models used. Additionally, maintaining the type of isotherm before and after fixing the CALB on the two MOFs confirms that their organo-metallic structure was not altered.

#### 2.1.4. Surface Morphology

The morphology of the MOFs before and after fixing CALB are shown in Figure 4. Figure 4a shows the free MOF-199 in MOF, Figure 4b shows the image corresponding to the CALB biocatalyst -MOF-199, Figure 4c shows the SEM of free ZIF-8, and, finally, Figure 4d shows the CALB-ZIF-8 biocatalyst.

Figure 4a,b show the images corresponding to free MOF-199 and the biocatalyst, CALB-MOF-199, respectively. Free MOF-199 clearly shows its typical double pyramid shape for SEM, consistent with the scientific literature. The average size of the crystals examined is between 2 and 5 nm. With CALB lipase adsorption, MOF-199 maintained its morphology, and enzyme binding did not affect the organo-metallic structure. Interestingly, the images show a very heterogeneous coverage of the CALB observed in certain conglomerate points (see Figure 4b). This is clear evidence that the Lipase was adsorbed on MOF-199. As shown in Figure 4c, ZIF-8 disperses as dodecahedral rhombus crystals with a particle size of a few nanometers, which coincides with what has already been reported in the specialized literature. The SEM image in Figure 4d indicates that the crystal morphology and apparent particle size of CALB-ZIF-8 are maintained upon CALB immobilization. However, a slight roughness is observed that makes it different from free ZIF-8, which is very clear evidence that there was physical adsorption of the CALB lipase on the surface of ZIF-8, and that this process does not destroy the basic structure of the ZIF-8. These results are consistent with what was previously found and analyzed using DRX and FTIR techniques.

### 2.2. Lipase Adsorption Isotherms

The analysis of how lipases from an aqueous solution approach MOFs and understanding of this phenomenon are very important in the development of biocatalysts and prompted us to address this phenomenon in the context of adsorption in our investigation. The moment when equilibrium is reached during adsorption between the CALB and MOF systems is understood to be the most important piece of data needed to correctly understand the adsorption process. Therefore, developing a model of the isotherms of the CALB-MOF system is important as it helps us form an understanding of the pathways of the adsorption mechanisms and can be used as an effective design to synthesize biocatalysts of this nature. In recent times, linear regression analysis has been one of the most applied tools to define the adsorption models that best fit as it quantifies the distribution of adsorbates, analyzes the adsorption system, and verifies the consistency of the theoretical assumptions of the model of each adsorption isotherm that is applied [36]. In this work, we model the results of the CALB adsorption on the MOFs using equations of two and three parameters to adequately interpret which model the system best fits and develop a better understanding of the mechanism used to adsorb the Lipase onto the MOFs.

The experimental results of CALB equilibrium adsorption on MOF-199 and ZIF-8 fitted from two-parameter (Freundlich, Langmuir) and three-parameter (Sips, Redlich-Peterson (RP), Radke-Prausnitz and Toth) models are shown in Figure 5a,b, respectively, presented only at 25 °C (Table 2 presents the summary of all the results obtained at the three temperatures tested). Each one of the parameters and the R^2^ correlation coefficient, which was used as a criterion to establish which model fits best and that describes the lipase adsorption on the MOFs, are presented in Table 2 for the two MOFs tested in this investigation. The experimental data *q_e_* (representing the lipase adsorption capacity on the MOF expressed as mg·g^−1^ of the respective MOF) and *C_e_* (representing the lipase concentration at equilibrium expressed as mg·L^−1^) were fitted according to the models mentioned and explained in detail in Section 3.2.1 to understand the types of interactions and the mechanisms of Lipase adsorption on the surface of each MOF. To establish each of the fit parameters of the models, the Rosenbrock and quasi-Newton optimization method included in the Statistica^®^ software v14 were used. Table 2 shows the parameters and coefficients obtained from the corresponding data for lipase adsorption, applying these models. The model that best describes the two biocatalysts was that of Sips, which was derived from the original model of the Langmuir isotherm, as described in Section 3.2.1 (where each of them was described in good detail, the models that were used in this investigation). This model is based on the premise that the adsorption process is presented in a monolayer throughout the mathematical development of said isotherm at high adsorbate concentrations for Langmuir-type modeling and is subsequently reduced to the form of the Freundlich isotherm model at low concentrations of adsorbate. As can be seen in the results reported in Table 2, the model that best fits the CALB-MOF-199 system at the different temperatures investigated is precisely that of Sips, considering that it presented the best R^2^ compared with the other models applied. From a global point of view of the results, it can be seen that the parameters of each model (except for the Redlich–Peterson model) applied to the CALB-MOF-199 biocatalyst have an acceptable coincidence factor (R^2^).

According to the results shown in Figure 5a,b, in addition to those reported in Table 2, the adsorption capacity of Lipase increases as a function of the temperature variable, a result that draws attention compared to other types of adsorbate–adsorbent interactions; however, this fact has already been reported in the literature [28]. This allows us to propose that when the temperature increases in the range studied, allowing the networks of the two MOFs to expand slightly, simultaneously with the CALB, due to the effect of the temperature, it allows their “spiral” to unfold and enter the interior of the pores more easily. Therefore, a greater amount of CALB adsorbed on the MOFs is generated. If the adsorption capacities of the biocatalysts synthesized in this work are compared according to the results applying the previously explained models, the maximum adsorption capacity of CALB on the MOFs is very similar between them according to the temperatures studied. When determining the values of the thermodynamic variables with the follow Equation (1):∆G° = ∆H° − T∆S°(1)
it was found that the Gibbs free energy (ΔG°), enthalpy (ΔH°) and entropy (ΔS°), which can be used to evaluate the spontaneity of the adsorption process [37]. The values obtained in this investigation were positive values of ΔG° at all the temperatures tested. This shows that the adsorption processes in the Lipase-MOF, in general, are not spontaneous, from these thermodynamic premises. The enthalpy values of adsorption, ΔH° (whose values are in the range of 43.56 and 287.32 kJ·mol^−1^) for the hydrophilic surfaces, ZIF-8 and MOF-199, resulted in positive values, which confirmed the endothermic nature of the process. When ΔH° is less than 50 kJ·mol^−1^, adsorption is considered physisorption [36,37,38], as in the case of CALB-ZIF-8, while adsorption was chemisorption at values greater than 50 kJ·mol^−1^ for CALB. -MOLF-199. The positive values for the entropic variable ΔS° for the CALB-MOF-199 and the CALB-ZIF-8 indicate the randomness in the solid–solution interface during the adsorption process, and a good affinity [28,37]. To explain these thermodynamic results, one must understand the forces of attraction. Adhesion of Lipase to ZIF-8 is by physical adsorption, while in the case of MOF-199, it is by chemical adsorption. In the first case, the attractive forces are mainly hydrophilic and Van der Waals forces, whereas they are chemical bonds in the second MOF. As the temperature increases, the adhesion of hydrophilic proteins to the hydrophilic surface of ZIF-8 compensates for the drop in Van der Waals forces [11,28,38]. However, in the case of MOF-199, the extent of chemisorption increased with increasing temperature, and enzyme binding increased [17] the adhesion of hydrophilic proteins to the hydrophilic surface of ZIF-8, which compensates for the drop in Van der Waals forces [11,28,38]. 

### 2.3. Adsorption Kinetics

To examine the mechanism of the CALB adsorption process on the two MOFs used in this research, several kinetic models have been used: pseudo first order (PFO), pseudo-second order (PSO), Elovich model, and Intraparticle. In the pseudo-first order model, the sorption rate is proportional to the adsorbate concentration. Figure 6 shows the kinetic profiles of adsorption of the biocatalysts CALB-MOF-199 and CALB-ZIF-8 at different concentrations and initial temperatures (from Figure 6a–f). As shown in the kinetic profiles, in general, there was a rapid initial absorption of Lipase from the solution, where after 30 min it began to stabilize and reach the equilibrium plateau after 90 min; this was measured during the monitoring time of the Lipase-MOF adsorption process. This rapid adsorption can be attributed, as mentioned in the literature [23,24,25,26,27,28], to the presence of many available empty adsorption sites [7,28]. As the Lipase adsorbs, the number of these available sites decreases, the slope flattens out, and the adsorption rate decreases [39].

It should be noted that this very rapid adsorption exceeds those found in the literature and gives further advantage to the results obtained here. In the case of the CALB-ZIF-8 biocatalyst (Figure 6d–f), there was a trend that at low lipase concentrations, the adsorption rate increased as a function of the initial concentration. However, the importance of this effect decreased at higher concentrations. This effect is widely explained in the literature, based on the driving forces of the concentration gradient, which becomes large enough so that any increase does not generate greater adsorption. It is worth bearing in mind that it is probable that if the lipase concentration is increased, an agglomeration phenomenon will be generated that goes against the speed of diffusion and the utilization capacity of the interior pores [6,9]. Here, it is verified again that as the temperature decreases, the adsorption of Lipase on the MOF’s decreases as well as at high concentrations. This decrease can be associated with the fact that the Van der Waals forces of attraction are low. Suppose the behavior of the CALB-MOF-199 biocatalyst is analyzed. In that case, it is possible to establish that there is a similar behavior concerning what has already been found with CALB-ZIF-8 concerning the concentration, where the speed of lipase adsorption (factor kinetic) increases with increasing concentration, but to a lesser extent.

The effect of temperature on CALB-MOF-199 was similar to that of ZIF-8, in which an increase in temperature increased the rate of adsorption. When investigating the kinetics of the biocatalysts, the results show that there is a great dependence between the physical and chemical characteristics of the biocatalysts, which means that they have a great influence on the adsorption mechanisms, which can eventually be through a film or diffusion into the pore network, or a combination of the two may occur [7,28,40,41,42]. To investigate the interaction mechanisms from the kinetics of lipase adsorption on the surface of the MOFs, these were adjusted to the following models: Pseudo-first order (PFO), Pseudo-second order (PSO), Elovich and Diffusion, and Intra-Particle (IPD) using the experimental results. The respective parameters (already presented in Section 3.2.2) and using the correlation factor (R^2^) as the adjustment criterion, where the kinetic model whose value is closest to 1, will be established as the one that gives a better explanation of the kinetic mechanism between CALB-MOF adsorption. The values corresponding to the parameters are found in Table 3 and Table 4. The results corresponding to the kinetics of PFO, PSO, and Elovich are shown in Table 3. From there, it is worth noting: (i) that for the PFO model, k_1_ presents an inverse behavior concerning the concentrations of lipases tested; that is, when the concentration was increased, the value of k_1_ decreased. Regarding temperature, Lipase, as a function of the temperature variable, increased in magnitude throughout the temperature study range; (ii) for the PSO kinetic model, the same phenomenon occurred; however, the magnitude is smaller for the k_s_. Some authors attribute this behavior to the tendency of Lipase to migrate from the solid phase to the bulk phase due to the effect of temperature [28,40]. It is worth mentioning that the R^2^ are not close to 1, so the adsorption of Lipase on the MOFs is not well described through either of these two kinetic models: PFO or PSO.

Table 3 shows the results adjusted to the Elovich kinetic model, which presents an R^2^ that allows us to infer that the results for the biocatalysts can be interpreted using this model. However, it should be noted that the Elovich equation does not predict any defined mechanism. Still, it is interesting if one wishes to describe an adsorption process on highly heterogeneous adsorbents [28,29], as is the case of MOFs, in particular used here: MOF-199 and ZIF-8, whose structure, as demonstrated during the textural analysis, corresponds to a heterogeneous porous network. If the R^2^ criterion is relaxed to examine the models, it can be seen, according to the R^2^ values obtained and reported in Table 3, that R^2^, without being a value very close to 1, can be considered an acceptable value as a fit for the CALB-MOF-199 biocatalyst for the pseudo-second order model. This fact may be showing that the lipase adsorption on the MOF-199 structure can occur through a surface level exchange until the functional sites on the surface are fully occupied; after this occurs, the lipase molecules begin to diffuse into the MOF-199 network where additional interactions (such as inclusion complexes, hydrogen bonding, hydrogen phobic interactions) probably occur [40]. The PSO model assumes that each lipase molecule adsorbs to two adsorption sites, allowing a stable binuclear bond to form [28] for the pseudo-second order model. This fact may demonstrate that the lipase adsorption on the MOF-199 structure can occur through a surface level exchange until the functional sites on the surface are fully occupied; after this occurs, the lipase molecules begin to diffuse into the MOF-199 network where additional interactions (such as inclusion complexes, hydrogen bonding, hydrogen phobic interactions) probably occur [40,41,42]. 

These kinetics results have not fit well with the models studied up until now and it was not possible to correctly describe the kinetics or analyze the speed control step in the solid–liquid adsorption process. For this reason, the IPD [7,28] was addressed and used in order to better understand the lipase diffusion mechanism on the MOF surfaces used in this investigation, the results of which are shown in Table 4, which were deduced in the respective graphs (which are not recorded in this study). The graphs for the Lipase-MOF adsorption process are multilinear, which means that one part represents the external resistance (the steepest graph and the initial part); the other graphic, which is a little softer, implies some control of intraparticle diffusion [29]. Under the conditions of this experiment, it was found that at high concentrations of Lipase, intraparticle diffusion had a very low starting point, some almost at zero, which allows us to deduce that lipase adsorption at higher concentrations was completely controlled by diffusion intraparticle. It was possible to establish that external mass transfer occurs at low concentration ranges [16,17,18,19,20,21,22,23,24,25,26,27,28]. Thus, it is shown that IPD was not the only rate-limiting step and film diffusion was also involved in the mechanism [7,28]. In summary, lipase adsorption on MOFs can proceed as follows: (i) transport of Lipase from the boundary film to the external surface of the adsorbent (film diffusion), (ii) transfer of the lipase molecule from the surface to the interior, or particular active sites, and (iii) uptake of Lipase by the surface-active sites ZIF-8 and MOF-199 [7,8,9,10,11,12,13,14,15,28]. In the initial stage of the adsorption process, film diffusion is an important step in rate control [10,11,12,13,14,15,28].

The intraparticle diffusion model suggested that mass transfer affects adsorption at lower concentrations [42,43]. The intraparticle diffusion constant, K_id_, was found to increase with enzyme concentration due to increasing driving force, i.e., the concentration gradient [28]. In the initial stage of the adsorption process, film diffusion is an important step in rate control [10,11,12,13,14,15,28]. The intraparticle diffusion model suggested that mass transfer affects adsorption at lower concentrations [42]. The intraparticle diffusion constant, K_id_, was found to increase with enzyme concentration due to increasing driving force, i.e., the concentration gradient. The values of C (nm^2^·min^−1^) for the biocatalyst CALB-MOF-199 at low concentrations oscillate between 0.0077 and 0.0176 and at high concentrations between 0.084 and 0.097, while for CALB-ZIF-8 at low concentrations, it ranged between 0.0321 and 0.0349, and at high concentrations, it varied between 0.00443 and 0.0487. This behavior is very similar to that reported and widely explained in the literature by other authors [28].

### 2.4. Transesterification Reaction and Operational Stability

When tested under the same experimental conditions using the biocatalysts prepared in this investigation (40 °C, methanol/oil ratio 12:1), the biodiesel yield obtained after 4 h of reaction was 93.4% and 83.5%, using the biocatalysts CALB-ZIF-8 and CALB-MOF-199, respectively. The highest yield was obtained for the CALB-ZIF-8 biocatalyst without the performance of the CALB-MOF-199 biocatalyst being considered low. It is worth noting that this performance is much higher than others reported with this type of system in the literature (LIPASES-MOF’s), so the systems reported here make them very promising biocatalysts to be used in the chemical industry sector. This result is supported favorably due to the interaction of hydrophobic characteristics, which prevents the biocatalyst’s deactivation in the case of contact with aqueous systems. The high performance must also be associated with the larger volume and pore radius of the ZIF-8 compared to the other MOFs tested.

Finally, for industrial purposes, it is necessary to study the stability and reuse of the biocatalyst, which are essential variables [4,5,6,27,28]. In this investigation, the biocatalysts, in terms of performance, gradually decreased each time they were re-used, which shows that Lipase is leached from the interior of the MOF’s structure [17,43]. Some tests were carried out as additional tests based on the reuse of the biocatalysts. From this point of view, the one that presented the best properties was the CALB-MOF-199, presenting a better performance and maintaining more than 80% of the initial activity up to five cycles, which are very severe conditions from an experimental point of view, making it a biocatalyst with excellent properties. This is because the adsorption on MOF-199 was chemical, which is stronger than when compared to the physical adsorption that occurred with ZIF-8, which prevented Lipase from leaching. However, despite its higher stability, the Lipase chemically adsorbed to the interior of MOF-199 had the lowest activity, a common problem with chemical immobilization. The Lipase adsorbed on ZIF-8 was the least stable, and the activity decreased to about 10% only in the fourth cycle of the five carried out, and it maintained an activity of 60% until the third cycle, which is a common problem with chemical immobilization. 

It is highlighted that in this work, n-hexane was used as the reaction medium, and the results show that the enzymatic reaction speed was increased [39,40,41,42,43]. A process of inhibition action of methanol is presented only if the amount placed in the reaction system is greater than its solubility. It has been reported by several authors that when methanol is separated from one of the phases in the reaction, an unfavorable unfolding of the enzyme occurs by stripping it of essential water molecules, which impairs its activity [25,26,27,28,29,43]. This is why adding an organic solvent to the reaction mixture increases solubility and reduces the inhibitory effect of methanol [26,27,28,29]. This action will result in the experimenter being able to use more alcohol.

## 3. Materials and Methods

### 3.1. Reagents

Candida Antarctica Lipase B (CALB), MOF-199 (Merck-Sigma Aldrich: Burlington, MA, USA), ZIF-8 (Merck-Sigma Aldrich), African palm oil (Colombian production), Phosphate Buffer Saline, and pH 7.4 (Sigma Aldrich: St. Louis, MO, USA) were obtained from Thermo Fisher Scientific, Waltham, MA, USA. Methanol of 99.5% purity (Fisher reagents), FAMES Merck Standard Mix, GC Analysis of a Grain Fatty Acid Methyl Ester (FAME) Mix on SP™-2560, (Merck-Sigma Aldrich: Burlington, MA, USA) (composition: 1. Caprylic (C8:0) 1.9 wt.%; 2. Capric (C10:0) 3.2 wt.%, 3. Laurie (C12:0) 4. Tridecanoic (C13:0) 3.2 wt.%, 5. Myristic (C14:0), 6. Myristoleic (C14:1n9c) 1.9 wt.%, 7. Pentadecanoic (C15:0) 1.9 wt.%, 8. Palmitic (C16:0) 13.0 wt.%, 9. Palmitoleic (C16:1n9c) 6.4 wt.%, 10. Heptadecanoic (C17:0) 3.2 wt.%, 11. Stearic (C18:0) 6.5 wt.%, 12. Elaidic (C18:1n9t) 2.6 wt.%, 13. Oleic (C18:1n9c) 19.6 wt.%, 14. Linoleic (C18:2n60) 13.0 wt.%, 15. Arachidic (C20:0) 1.9 wt.%, 16. cis-11-Eicosenoic (C20:1) 1.9 wt.%, 17. Linolenic (C18:3n3) 6.4 wt.%, 18. Behenic (C22:0) 1.9 wt.%, 19. Erucic (C22:1n9) 1.9 wt.%), Sigma Aldrich Deionized Water (CAS Number:7732-18-5).

### 3.2. Batch Adsorption of Lipase onto Activated MOFs

To prevent the structures of the MOFs used from being modified during the experiments through contact with the environment, they were subjected to a pre-treatment procedure consisting of bringing them to an initial temperature of 60 °C and a vacuum of 10^−5^ mbar. Subsequently, MOF-199 was activated at 125 °C for 6 h, while ZIF-8 was activated at 110 °C for 8 h. It should be noted that due to the heat treatment of MOF-199 to activate it, its color changed from light turquoise to dark blue (characteristic of copper, which is a metal in the center of the organic metal lattice). Before running the batch adsorption experiments, 0.23 g of the respective activated MOFs were taken, and 10 mL of phosphate buffer solution (pH 7.6) was added to the sealed glass vials [28]. The MOFs were then dispersed by sonication at an amplitude of 50 Hz for 15 min; then, the enzyme (CALB) was added to each. Next, 10 mL of the respective CALB Lipase solutions, at different concentrations, were added to the sealed vials and placed in a constantly stirred water bath at 300 rpm and different temperatures from 25 °C to 40 °C. The initial concentrations of enzymes in the mixtures ranged between 3.45 and 0.8570 mg·mL^−1^ [28]. At regular intervals, 25 μL of the sample was withdrawn and filtered. Next, 200 μL of Bradford’s reagent [29] was added, and the optical density was measured at 595 nm using a microplate UV spectrophotometer (Multiskan GO, Leicestershire, UK). The enzyme concentration in the extracted samples was determined by comparing the optical density to a calibration curve prepared using serial dilutions of standard protein, albumin, and solution of known concentration. After 48 h, MOFs with immobilized Lipase (CALB) were collected by centrifugation at 9000 rpm and −4 °C for 15 min and washed twice with buffer solution [28]. The concentration of enzyme (CALB) remaining in the supernatant after 48 h was assumed to be an equilibrium concentration [28,29]. The respective MOF with the adsorbed Lipase, now called biocatalysts in this investigation, CALB-MOF-199 and CALB-ZIF-8, were lyophilized for further use. Enzyme immobilization efficiency (EIE) and immobilization capacity (*q_m_*) were determined using Equations (2) and (3), respectively [25,28]:(2)$$EIR=\frac{\left(Ci-Cf\right)}{Ci}\times 100\%$$
(3)$$q_{m}=\frac{V\left(Ci-Cf\right)}{m}$$
where *Ci* and *Cf* (mg·mL^−1^) are the initial and final concentrations of the enzyme (CALB), respectively, *q_m_* (mg·g^−1^) is the capacity of the MOF, *m* (g) is the weight of the MOF, and *V* (mL) is the volume of the solution.

To evaluate the mechanism of CALB adsorption on MOF-199 and ZIF-8, experimental equilibrium adsorption data were interpreted using the Freundlich, Langmuir, Sips, Redlich-Peterson (RP), Radke-Prausnitz, and Toth adsorption isotherm models. The isotherm constants were determined by measuring the equilibrium amount of CALB adsorption on the MOF (*q_e_* (mg·g^−1^)) and the equilibrium aqueous concentration of the protein (*C_e_* (mg·mL^−1^)) and adjusting them to the linear form of the isotherm models. All the experiments were repeated three times, and the mean and average values were considered. The standard deviation of the mean values of the triple tests in all the experiments was less than 2%.

#### 3.2.1. Adsorption Models

In this research, different models have been proposed to explain the adsorption process, adjusting the experimental data of the isotherms to these models, to explain the mechanism of lipase adsorption on the two MOFs under investigation in this study. Table 5 shows the models used to analyze the adsorption results.

#### 3.2.2. Adsorption Kinetics Models

Kinetic studies make it possible to describe an adsorbate’s adsorption rate on the corresponding adsorbent and determine the time at which maximum saturation is reached. Table 6 shows the most widely used kinetic models.

### 3.3. Biodiesel Production

To carry out the transesterification experiments with African palm oil using the biocatalysts and synthesize the biodiesel, the CALB was immobilized on the MOFs under the optimal conditions found with the experiments carried out in Section 2.2. The reaction mixture consisted of 3.5 g of African palm oil and 2.00 mL of methanol (equivalent to a 1:12 molar ratio), and 2 mL of n-hexane, as a preliminary test as reported in the literature [28,56,62]. When this test was carried out with the enzyme, 0.2 g of CALB was added to start the reaction. On the other hand, when the soluble free enzyme was assayed, 2.5 mL of the enzyme solution were used. This reaction mixture was treated and duly monitored under a temperature of 45 °C and constant stirring at 115 rpm using a thermostat. The reaction was allowed to proceed for 8 h to ensure a complete reaction and then a non-polar solvent was added to extract the biodiesel produced. In this experiment, n-hexane was used in an amount of 8 mL.

As mentioned in the introduction to this paper, additional experiments were carried out at the end of each reaction (those previously described) in order to make comparisons with the literature [28] and emphasize the importance of analyzing the reuse of immobilized CALB in both MOF-199 and ZIF-8. The immobilized enzyme was separated by centrifugation at 9500 rpm and reused in other runs using fresh African palm oil, methanol, and hexane. This procedure was repeated for six runs, and the FAMEs produced for each run were determined.

Transesterification studies for biodiesel synthesis, including diffusion studies, found in the literature are interesting. A study of diffusion in CALB adsorbed on MOFs MOF-199 and ZIF-8 was mathematically modeled in this work using different initial concentrations of African palm oil, keeping methanol and lipase concentrations constant; the volume of n-hexane was adjusted to bring the total volume to 10 mL [27,28]. The methanol concentration and Lipase loading in this investigation were not changed during the experiments; their concentrations were 0.078 mol·mL^−1^ and 0.05 mg·mL^−1^, and the concentration of African palm oil changed in the range from 0.85 to 0.45 mg·mL^−1^. In the specialized literature, it has been assumed that biodiesel production is linear during the first 4 h; according to our experience it is preferable to assume this for 8 h, as we have found that linearity is not always reached in 4 h, and this can affect the final analysis of the results. The initial reaction rate in each condition was calculated by dividing the concentration of FAMEs after 8 h of reaction by 8 h. 

### 3.4. Gas Chromatography Analysis

A 1 mL sample of the upper layer of n-hexane was extracted and sent to gas chromatography (GC) (Agilent 8860 Gas Chromatograph, Santa Clara, CA, USA) for analysis of fatty acid methyl esters (FAME’s). The design of the experiment using gas chromatography (GC) was as follows: the injector was operated at 250 °C split/splitness, split ratio 100:1; column: DB-Fst FAME, 30 m × 0.25 mm, 0.25 µm; Carrier gas: Helium, 19 psi, constant pressure; oven: 80 °C (0.25 min), then 40 °C/min up to 165 °C (1 min), then 4 °C·min^−1^ up to 230 °C (4 min); FID: 260 °C; hydrogen: 40 mL·min^−1^; air: 400 mL·min^−1^ auxiliary gas (N_2_): 25 mL·min^−1^; injection: 1 µL. A 1 µL sample was injected into the column through a 0. 45 mm filter and compared to a calibration determined using a standard mix of FAMEs. The FAMEs production yield was determined using the following Equation (4) [28,45].
(4)$$FAME\;yield=\frac{m_{FAME}}{m_{oil}}\%$$
FAMEs and moil are the area weights of the FAMEs produced and the oil used, respectively. The experiments were performed in triplicate.

### 3.5. MOF Characterization

In this investigation, MOF-199 and ZIF-8 were analyzed using different techniques before and after the adsorption of the CALB enzyme. One of the techniques used was X-ray diffraction, which was carried out using a Rigaku Miniflex 300/660 X-ray diffractometer (XRD) (Japan), using a Cu Kα of wavelength λ = 1.54056 Å as the source of X-rays at a generator voltage of 40 kV and a generator current of 40 mA. The FTIR technique was also used to characterize the structure of the biocatalysts synthesized in this research; for this, the respective infrared spectra were obtained utilizing Fourier transforms (Nicolet iS50 FTIR Spectrometer, Thermo Fisher Scientific, Waltham, MA, USA).

The surface area (S_BET_) was determined using the Brunauer-Emmet-Teller (BET) equation, and the total pore volume (V_total_) was calculated from the amount of N_2_ adsorbed at a relative pressure of P/P^0^ = 0.99. The size distribution of the mesopores was obtained using the DFT (Density Functional Theory) method. Before the analysis, the samples were degassed at 300 °C for 8 h. A Quantachrome sortometer, IQ2 (Boynton Beach, FL, USA), was used for this analysis. The hydrophobicity of MOF-199 and ZIF-8 was determined by the immersion of calorimetry measurements in benzene and water using a homemade immersion calorimeter.

Enzymes and proteins, in general, are sensitive to temperature and tend to have relatively low optimum operating temperatures in the range of 40–75 °C. The Lipase used in this work, Candida Antarctica Lipase B, maintains its activity at temperatures reaching 85 °C; at higher temperatures, the enzyme is denatured and inactivated [46]. As the two MOFs used in this work have a high thermal stability—higher than 600 °C (greater than that of the enzyme)—they can be used as supports in the transesterification reactions of palm oil, especially as enzyme supports, and become what will later be biocatalysts [28,47,48,49]. After immobilizing the enzyme, Lipase-MOF supports were dried using the lyophilization process, a common technique used to dry heat-sensitive compounds, including proteins [47]. After performing the immobilization of the enzyme by adsorption on each of the MOFs, additional determinations were made to confirm this. For this, techniques such as SEM, XRD, and FTIR were used, among others.

## 4. Conclusions

The use of metal-organic frameworks (MOFs) as supports for immobilizing enzymes is very attractive due to their large surface area and pore size. The mechanism, kinetics, and thermodynamics of lipase immobilization were studied in detail by adsorption analysis on ZIF-8 and MOF-199, and the results were compared by encapsulation immobilization. It was shown that the adsorption equilibrium was best described by the Sips model, which was used to determine the thermodynamic properties. On the other hand, the results of the adsorption kinetics that best describe biocatalysts are due to the Elovich one, which corresponds to physical adsorption, and the pseudo-second-order model corresponding to chemical adsorption in CALB-MOF-199. It was found that the limiting step of the adsorption process is influenced by intraparticle diffusion for both biocatalysts. The results found in this work allow us to advance our deep understanding of the immobilization of lipases on MOFs, which is essential to analyze the potential application of these systems. In addition, the activity of the biocatalysts produced in the synthesis of biodiesel by the transesterification of African palm oil using n-hexane as a solvent medium was tested. The activity was high for all the biocatalysts prepared, and the stability was maintained during three reuse cycles. The highest biodiesel yield, 90.5%, was obtained using the CALB-ZIF-8, while the Lipase fixed by chemical adsorption on the MOF-199 showed the highest operational stability. In summary, this work’s results can potentially improve biodiesel production using Lipase immobilized on metal-organic frameworks, help us form an understanding of the mechanism of the adsorption reaction, and predict the kinetics of diffusion-reaction systems.

## Figures and Tables

**Figure 1 ijms-24-10741-f001:**
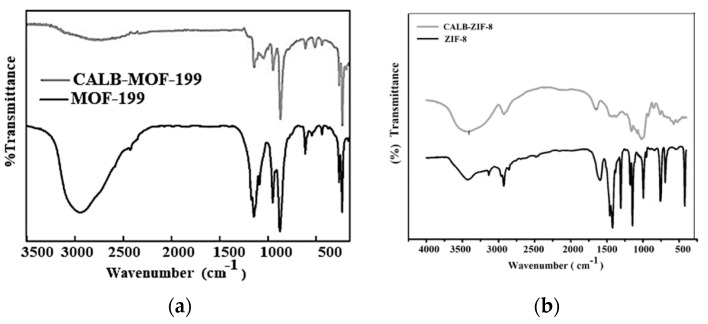
FTIR corresponding to free MOFs: (**a**) Free MOF-199; (**b**) CALB-MOF-199.

**Figure 2 ijms-24-10741-f002:**
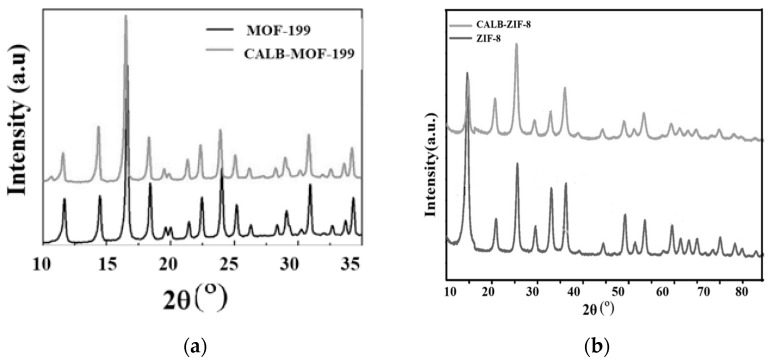
DRX corresponding to MOFs: (**a**) Free, Black-MOF-199 and Grey, CALB-MOF-199; (**b**) Free, ZIF-8 and Grey, CALB-ZIF-8.

**Figure 3 ijms-24-10741-f003:**
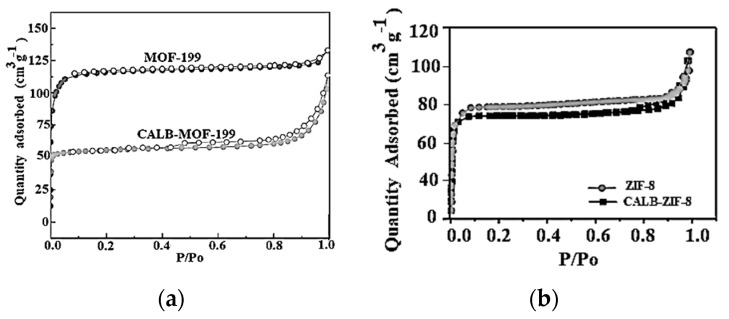
The N_2_ adsorption–desorption isotherms: (**a**) MOF-199 free and impregnated with Lipase Candida Antarctica (CALB-MOF-199); (**b**) ZIF-8 free and impregnated with Lipase Candida Antarctica (CALB-ZIF-8).

**Figure 4 ijms-24-10741-f004:**
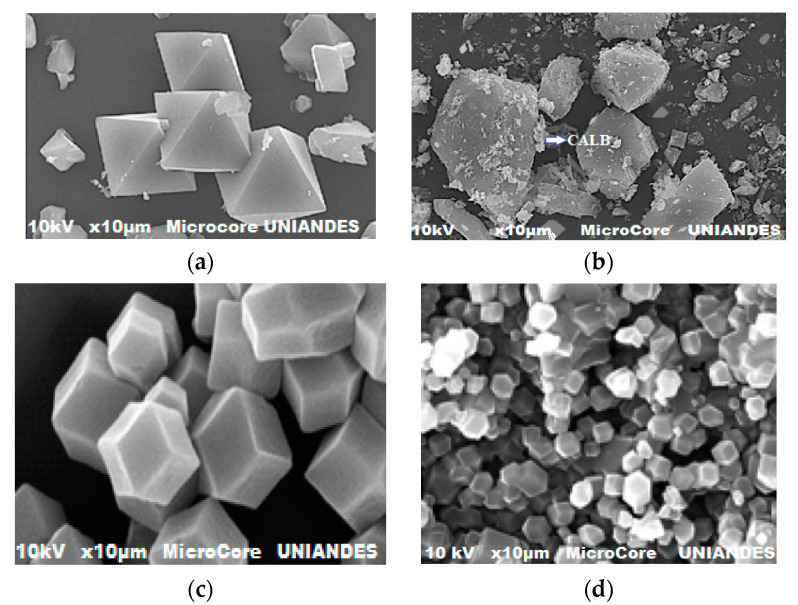
SEM correspond to samples free and impregnated: (**a**) MOF-199; (**b**) CALB-MOF-199; (**c**) ZIF-8; (**d**) CALB-ZIF-8.

**Figure 5 ijms-24-10741-f005:**
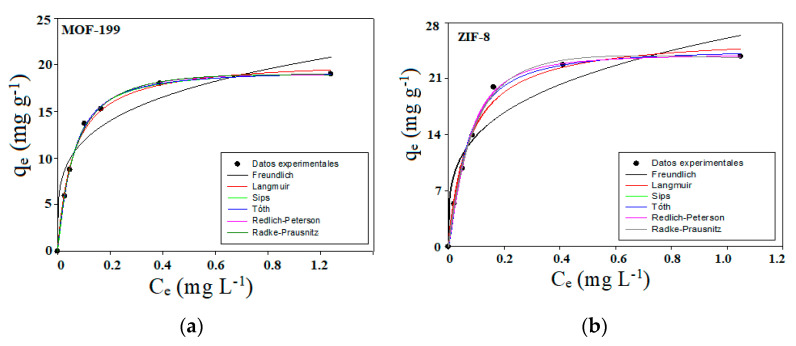
Comparison of experimental data and adsorption isotherms predicted for Lipase CALB on (**a**) MOF-199 and (**b**) ZIF-8: Adjusted to Freundlich, Langmuir, Sips, Redlich-Peterson (RP), Radke-Prausnitz and Toth. models, for the adsorption of CALB on MOF-199 and ZIF-8.

**Figure 6 ijms-24-10741-f006:**
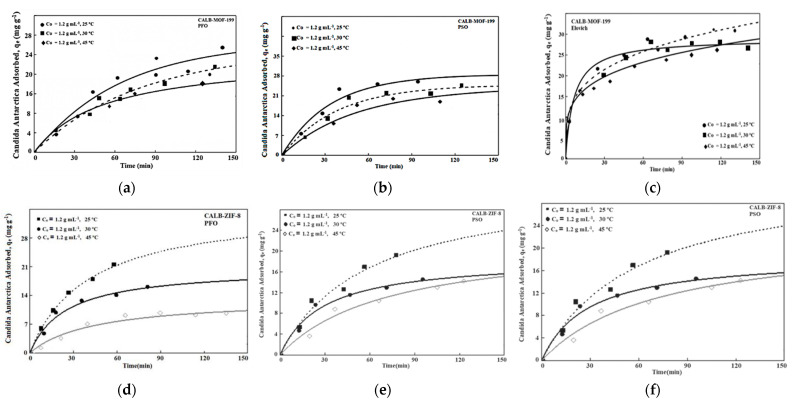
Results at fixed concentration at different temperatures for the Lipase-MOF system, at different temperatures: Adjusted to PFO, PSO and Elovich.

**Table 1 ijms-24-10741-t001:** DA and DFT parameters obtained from the adsorption-desorption isotherm of N_2_ at 77.4 K.

Samples	S_BET_ [m^2^·g^−1^]	DA (P/P^0^ < 0.1)				DFT (P/P^0^ 10^−7^ − 1)	
		V_mic_ [cm^3^·g^−1^]	E_o_ [kJ·mol^−1^]	n	Pore Radius [Å]	V_P_ [cm^3^·g^−1^]	Half PoreWidth [Å]
MOF-199	1750	0.66	8.47	3.4	7.0	0.69	3.52
CAL-MOF-199	1400	0.54	8.21	3.1	5.5	0.55	2.70
ZIF-8	1733	0.88	4.47	3.4	8.6	0.65	4.63
CALB-ZIF-8	1350	0.73	4.12	2.9	7.2	0.59	4.00

**Table 2 ijms-24-10741-t002:** Adjustment parameters to the models of the Freundlich, Langmuir, Sips, Redlich-Peterson (RP), Radke-Prausnitz and Toth, for the adsorption of CALB on MOF-199 and ZIF-8.

Model	Parameters	CALB-ZIF-8		
		25 °C	30 °C	40 °C
Langmuir	q_max_ (mg·g^−1^)	27.64	32.12	36.78
	K_L_ (mg^−1^·g^−1^)	7875	8034	8934
	R^2^	0.9789	0.9789	0.9834
	R_L_	0.015	0.019	0.021
Freundlich	K_F_ (mg·g^−1^) (L·mg^−1^) 1/n	6845	7987	8005
	1/n	0.301	0.312	0.379
	R^2^	0.9843	0.9834	0.9856
Sips	q_max_ (mg·g^−1^)	30.79	38.57	41.07
	K_s_ (L·mg^−1^)	0.3487	0.2394	0.2065
	n_s_	0.8943	0.8631	0.8304
	R^2^	0.9993	0.9989	0.9985
Redlich-Peterson (RP)	K_R_ (L·g^−1^)	0.894	1765	1898
	a_R_ (L·mg^−1^)	0.398	0.299	0.272
	β	0.865	0.887	0.898
	R^2^	0.9587	0.9698	0.9456
Radke Prausnitz	q_mRP_ (mg·g^−1^)	27.98	29.67	30.73
	K_RP_ (L·mg^−1^)	0.078	0.086	0.094
	m_RP_	1087	1435	1754
	R^2^	0.9876	0.9798	0.9895
Toth	q_mT_ (mg·g^−1^)	28.54	33.88	35.67
	K_T_	0.354	0.185	0.234
	m_T_	0.786	0.804	0.847
	R^2^	0.9876	0.9910	0.9854

**Table 3 ijms-24-10741-t003:** Adsorption kinetic parameters obtained by the PFO, PSO, and Elovich models.

T (°C)	C_o_	q_e_ (mg·g^−1^)	Pseudo-First Order		Pseudo-Second Order		Elovich Model		
			k_1_ (min^−1^)	R^2^	k_2_ (g/mg·min)	R^2^	*α* (m/g/min)	1/*β* (mg/g)	R^2^
CALB-MOF-199									
25	1.2	21.34	0.0165	0.865	0.0087	0.916	3.456	2.154	0.967
	0.9	18.89	0.0189	0.856	0.0122	0.903	3.165	2.078	0.974
	0.6	16.56	0.0216	0.810	0.0146	0.892	2.678	1.896	0.962
	0.3	15.74	0.0365	0.845	0.0245	0.898	2.345	1.654	0.981
30	1.2	23.34	0.0254	0.798	0.0076	0.912	3.765	3.154	0.943
	0.9	19.65	0.0278	0.795	0.0116	0.943	3.376	2.986	0.967
	0.6	18.67	0.0306	0.795	0.0116	0.943	3.376	2.986	0.967
	0.3	17.56	0.0389	0.812	0.0213	0.904	2.896	1.967	0.987
40	1.2	26.34	0.0345	0.797	0.0063	0.934	5.875	4.034	0.978
	0.9	21.96	0.0374	0.807	0.0104	0.921	4.965	3.871	0.985
	0.6	18.26	0.0387	0.804	0.0124	0.912	4.762	2.987	0.976
	0.3	17.32	0.0402	0.808	0.0202	0.899	3.986	2.487	0.956
CALB-ZIF-8									
25	1.2	19.45	0.0168	0.832	0.0058	0.934	3.876	2.376	0.919
	0.9	17.87	0.0176	0.834	0.158	0.925	3.653	2.267	0.921
	0.6	15.12	0.0201	0.810	0.0187	0.932	3.521	2.056	0.937
	0.3	14.76	0.0304	0.786	0.0289	0.912	3.312	1.965	0.934
30	1.2	22.52	0.0215	0.734	0.0046	0.906	3.985	3.452	0.923
	0.9	18.31	0.0287	0.782	0.0164	0.909	4.763	3.296	0.965
	0.6	17.10	0.0300	0.796	0.0174	0.934	4.965	3.038	0.965
	0.3	16.23	0.0321	0.895	0.0265	0.901	5.098	2.896	0.976
40	1.2	23.33	0.275	0.876	0.0032	0.921	6.231	4.342	0.934
	0.9	20.05	0.0295	0.854	0.0153	0.902	5.865	4.106	0.976
	0.6	19.56	0.0312	0.845	0.0166	0.943	5.321	3.892	0.947
	0.3	18.45	0.0355	0.807	0.0243	0.931	5.031	3.753	0.967

**Table 4 ijms-24-10741-t004:** Parameters obtained from Intra-Particle Diffusion kinetic models (IPD).

T (°C)	C_o_	q_e_ (mg·g^−1^)	Intraparticle Diffusion (IPD)			
			K_pi1_	K_pi2_	C_i_ (nm^2^·min^−1^)	R^2^
CALB-MOF-199						
25	1.2	21.34	12.98	10.67	0.0076	0.997
	0.9	18.89	10.64	9.65	0.0087	0.994
	0.6	16.56	9.96	8.45	0.0245	0.998
	0.3	15.74	9.65	7.85	0.0287	0.994
30	1.2	23.34	13.98	12.92	0.0121	0.992
	0.9	19.65	11.43	11.65	0.0189	0.998
	0.6	18.67	10.07	9.87	0.0256	0.998
	0.3	17.56	9.87	8.94	0.0298	0.999
40	1.2	22.76	14.43	13.32	0.0132	0.994
	0.9	21.96	13.21	12.02	0.0143	0.992
	0.6	18.26	12.76	11.98	0.0321	0.997
	0.3	17.56	11.07	10.65	0.0346	0.998
CALB-ZIF-8						
25	1.2	19.45	13.12	12.34	0.0084	0.997
	0.9	17.87	12.32	11.64	0.0097	0.996
	0.6	15.12	11.45	10.65	0.0256	0.994
	0.3	14.76	11.02	9.87	0.0297	0.993
30	1.2	22.52	14.21	13.76	0.0136	0.997
	0.9	18.31	13.77	12.86	0.0198	0.994
	0.6	17.10	12.87	11.98	0.0287	0.993
	0.3	16.23	12.22	10.76	0.0318	0.996
40	1.2	23.33	14.87	14.34	0.0146	0.997
	0.9	20.05	13.03	13.21	0.0177	0.993
	0.6	19.56	11.82	12.54	0.0443	0.996
	0.3	18.45	11.02	11.43	0.0487	0.997

**Table 5 ijms-24-10741-t005:** Adsorption models used in this work of two and three parameters.

Adsorption Models	Description
Langmuir	$$qe=q_{max}*\frac{K_{L}C_{e}}{{1+K}_{L}C_{e}}\;\frac{C_{e}}{q_{e}}=\frac{1}{q_{max}}C_{e}+\frac{1}{K_{L}q_{max}}$$ This isotherm is based on three assumptions: adsorption is limited to the monolayer coverage, all surface sites are equal, and the ability of a molecule to be adsorbed at a given site is independent of its occupancy of neighboring sites [44,45,46].
Freundilch	$$qe=K_{F}C_{e}^{1/n}\;logq_{e}={\log K}_{F}+\frac{1}{n}logC_{e}$$ The Freundlich Isotherm is a widely used empirical equation for describing adsorption equilibrium. The plot of log *q_e_* against log *C_e_* has a slope with the value of 1/*n*, and the intercept is *K_F_*. log *K_F_* is equivalent to log *q_e_* when *C_e_* = 1. However, in another case, when 1/*n*, the KF value depends on the units in which *q_e_* and *C_e_* are expressed ≠1. On average, a favorable adsorption Freundlich constant, *n*, is between 1 and 10. Increasing *n* implies a greater interaction between adsorbate and adsorbent, while 1/*n* = 1 indicates linear adsorption leading to higher adsorption energies, identical for all sites [47,48,49].
Toth	qe=qmax∗bTCe(1+(BTCe)nT)1nT This isotherm is derived from the potential theory. The Toth equation has proven to be a valuable tool in describing adsorption for heterogeneous systems. A quasi-Gaussian asymmetric energy distribution with the left side broadened is assumed, i.e., most adsorption sites have less energy than the mean value [50,51,52,53].
Redlich-Peterson	$$qe=\frac{K_{RP}C_{e}}{{1+a}_{RP}C_{e}^{\beta }}$$ Redlich-Paterson is an empirical equation, designated as the “three parameter equation”, capable of representing adsorption at equilibrium over a wide range of concentrations. Redlich and Peterson incorporate the features of the Langmuir and Freundlich isotherms into a single equation. Frequently applied in homogeneous or heterogeneous adsorption processes. There are two limiting behaviors, i.e., the Langmuir form and Henry’s law form [54]. β=1, β=0
Sips	$$q_{e}=q_{m}*\frac{K_{s}C_{e}^{\beta s}}{a_{s}C_{e}^{\beta s}}\;\beta_{s}lnC_{e}=-ln\left(\frac{K_{s}}{q_{e}}\right)+ln\left(a_{s}\right)$$ The Sips isotherm combines the Langmuir and Freundlich isotherms and is given the above general equation (left-hand side). Here *K_s_* is the constant of the Sips isotherm model (L·g^−1^), 𝛽_𝑠_ is the Sips isotherm exponent, and 𝑎𝑠 is the constant of the Sips isotherm model (L·g^−1^). The above right-hand paragraph also gives the linearized form [55]. This model is suitable for predicting adsorption on heterogeneous surfaces, thus avoiding the limitation that normally occurs during increasing adsorbate concentration, normally associated with the Freundlich model [55]. Therefore, this model reduces to the Freundlich model at low adsorbate concentrations, but at high adsorbate concentrations it predicts the Langmuir model (monolayer adsorption). The parameters of the Sips isotherm are a function of the pH, temperature, and concentration +, and isotherm constants differ by linearization and non-linear regression [55].
Radke Prausnitz	$$qe\frac{q_{MRP}K_{RP}C_{e}}{{\left(1+K_{PR}C_{e}\right)}^{MRP}}$$ The Radke-Prausnitz isotherm model has several important properties that make it very useful in adsorption at low adsorbate concentrations [56]. The previous expression gives the isotherm. In this equation, *q_mrp_* is the maximum Radke-Prausnitz adsorption capacity (mg·g^−1^), *K_PR_* is the Radke-Prausnitz equilibrium constant, and MRP is the exponent of the Radke-Prausnitz model. At a low adsorbate concentration, this model isotherm reduces to a linear isotherm, while at a high adsorbate concentration, it becomes the Freundlich isotherm, and when *M_PR_* = 0, it becomes the Langmuir isotherm. Another important feature of this isotherm is that it fits a wide range of adsorbate concentrations well. In this Radke-Prausnitz model, the model parameters are obtained by non-linear statistical fitting of experimental data [57].

**Table 6 ijms-24-10741-t006:** Adsorption Kinetics Models.

Kinetic Models	Description
Pseudo-First Order (Lagergren’s model)	The model given by Langergren is defined as $$\frac{dq}{dt}=k_{1}(q_{e}-q)$$ Integrating the equation concerning the boundary conditions *q* = 0 at = 0 and *q* = *q_e_* at = *t*, we obtain $$\log\left(q_{e}-q\right)=\log\left(q_{e}\right)-\frac{k_{1}}{2303}t$$ where *k*_1_ is the Lagergren adsorption rate constant (min^−1^); *q_t_* and *q_e_* are the amounts adsorbed at a time t and equilibrium, respectively, *t* in (min). The plot of log(*q_e_* − *q_t_*) as a function of time; the intercept is log *q_e_* and the slope is *k*_1_ [58].
Pseudo-Second Order	The pseudo second order equation based on equilibrium adsorption is expressed as: $$\frac{dq}{dt}=\;k_{2}{\left(q_{e}-q\right)}^{2}$$ Separating the variables in the above equation, we obtain $$\frac{dq}{{\left(q_{e}-q\right)}^{2}}=\;k_{2}dt$$ Integrating this equation with respect to the boundary conditions *q* = 0 at = 0 and *q* = *q_e_* at = *t*, we obtain: $$\frac{t}{q}=\;\frac{1}{k_{2}q_{e}^{2}}+\frac{1}{q_{e}}t$$ where *k*_2_ is the pseudo second order rate constant (g·mg^−1^·min^−1^); *q_t_* and *q_e_* are the amounts adsorbed at time t and equilibrium, respectively. The line graph of *t*/*q_t_* as a function of time has 1/*q_e_* as the slope and 1/*k*_2_ as the intercept. This rate constant is used to calculate the initial adsorption rate, h (mg·g^−1^·min^−1^), where *q_e_* is the equilibrium adsorption capacity, *k*_2_ (mg·g^−1^·min^−1^) is determined experimentally from the slope e intercept of the *t*/*q* plot versus *t* [59]. $q_{e}^{2}$
Intraparticle model (Weber-Morris)	Kinetic models do not identify the diffusion mechanism. The intraparticle diffusion model based on the theory proposed by Weber and Morris establishes a common empirical relationship in most adsorption processes since it varies proportionally with *t*^1/2^ more than with the contact time *t*. According to this theory, we have: $$q_{t}=k_{pi}t^{1/2}+C_{i}$$ where *k_pi_* (mg·g^−1^·min^−1/2^), the speed parameter for each stage, is obtained from the line *q_t_* versus *t*^1/2^ slope. *C_i_* is the intercept of stage *i*, giving an idea of the thickness of the boundary layer. If intraparticle diffusion occurs, *q_t_* versus *t*^1/2^ will be linear; if the graph passes through the origin, then the rate-limiting process is only due to intraparticle diffusion. Otherwise, another mechanism is involved along with intraparticle diffusion. In intraparticle diffusion plots, stage I is due to flash adsorption or external surface adsorption, where the adsorbate travels to the external surface of the adsorbent. In stage II, a gradual adsorption occurs where intraparticle diffusion is the rate limiting; that is, the adsorbate travels inside the pores of the adsorbent. In some cases, a stage III represents the final equilibrium where the intraparticle diffusion begins to decrease due to the low concentration of adsorbate; adsorption occurs inside the adsorbent [60].
Elovich model	This model is useful to understand chemisorption in an adsorption process (developed by Zeldowitsch). It makes it possible to predict the diffusion of mass and surface, a system’s activation, and deactivation energy. Although this model was initially used only for gaseous systems, its use was later extended to processes in aqueous solutions. The model assumes that the solute adsorption rate decreases exponentially as the amount of solute adsorbed increases. $$\frac{{dq}_{t}}{dt}=\alpha {exp}^{-\beta qt}$$ Since *q_t_* ≈ 0, ≈ *α* is the initial adsorption rate (mg/g·min), and *β* is the desorption constant. Integrating and applying the limits for *t* (0, *t*) and *q_t_* 0; *q_t_*, the Elovich model can be linearized as: $\frac{{dq}_{t}}{dt}$ $$qt=\frac{1}{\beta }ln\left[t+\frac{1}{\alpha \beta }\right]-\frac{1}{\beta }ln\left(\alpha \beta \right)$$ when the system approaches equilibrium, *t* ≫ 1/*αβ*, the previous equation becomes: $$q_{t}=\frac{1}{\beta }ln\left[\alpha \beta \right]+\frac{1}{\beta }lnt$$ The graph of *q_t_* versus at will help to establish the nature of the adsorption on the heterogeneous surface of the adsorbent, whether it is chemisorption or not [61].

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
