# Peer review of "Biodiesel Production Using Palm Oil with a MOF-Lipase B Biocatalyst from Candida Antarctica: A Kinetic and Thermodynamic Study"

_ijms, 2023, doi:10.3390/ijms241310741_

Round 1

Reviewer 1 Report

L. Giraldo et al. reported a full paper on kinetic and thermodynamic study of biodiesel production using combination of MOF and palm oil. Several models like Langmuir, Sips, etc has been used to develop adjusting parameters. This paper contains detailed research, however, several issues should be addressed below:

1. Line 502,503: the figure is not justified and should be labeled with a figure No.

2. Line 703 and 704: In the equation between line 703 and 704, for Ci and Cf, i and f should be subscripted. 

Once these were fixed, acceptance is recommended.

Author Response

May 31 2023

Dears
Ms. Maeve Wang
IJMC
MPDI

A greeting and and thank you for your contributions to this writing.
(1) too large Introduction part: we have greatly reduced the intruction now

(2) the lack of Figure caption on p.12: now the figure has its title

(3) inappropriate repeats in section 3.3, lines 740-747: This part has been corrected and all repeats have been removed.

Thank you very much and we will be attentive to your final decision.

Best Regards, 

Prof. Dr. Juan Carlos Moreno-Pirajan 
Full Professor
University of the Andes
Bogota, Colombia
On behalf of the authors

Reviewer 2 Report

The manuscript describes efforts at obtaining biodiesel by action of Candida antarctica Lipase B. The problem is actual, the experimental work appears to be suitable and the results are correctly interpreted (with one exception). The text, however needs stylistic and English corrections. This Reviewer suggests the addition of two relevant recent references.

List of comments/corrections:

(1) Abbreviations as MOF or ZIF should be explained where they appear at the first time in the manuscript. The use of un-explained abbreviations should be particularly avoided in the Title and in the Abstract.

(2) row 18: "better" - better than what?

(3) Figure 1. Legend for the right-side diagram is lacking. This makes confusion also in rows 223/224.

(4) rows 207 to 231:The sharp and intense IR bands in Figure 1 at 1500 to 700 and 1500 to 500 cm-1 should be commented on. (Or: at least an attempt should be made at the interpretation. The present text is confused.)

(5) in rows 555 and 714 is lacking a word?

(6) row 844: "funding aquisition" - it is strange to name this activity.

(7) row 877: 2020 should be 2020

(8) The following two recent references should be added:

Sun, X., Liu, S., Manickam, S., Tao, Y., Yoon, J. Y., Xuan, X.: Intensification of biodiesel production by cavitation. A review. Renewable and Sustainable Energy Reviews 2023, 179, 113277.

Diaz, L., Horstmann, F., Brito, A., González, L. A.: A comprehensive review of the influence of co-solvents on the catalysed methanolysis process to obtain biodiesel. Heliyon 2023, 9, E13006.

See above. No additional remarks.

Author Response

(The authors gave the same response as above.)

Reviewer 3 Report

Review attached.

.

Author Response

(The authors gave the same response as above.)

Round 2

Reviewer 3 Report

Unfortunately, the authors of the work did not comply with all the comments of the reviewers. Please, once again, carefully read the notes and make corrections in the manuscript.

.

Author Response

Response to Reviewer 3 Comments

Point 1: too large Introduction part

Response 1: we have greatly reduced the intruction now ( highlighted in yellow)

Point 2: the lack of Figure caption on p.12.

Response 2: The caption in Figure 6 (page 12) (highlighted in yellow) has now been placed in the Figure.

Point 3: Please check that all references are relevant to the contents of the manuscript.
Response 3: The references have been carefully checked and it is good for the reader of this topic to keep them in this paper in case he/she wishes to expand on any aspect.

Point 4. Minor revisions of English:

Response 4: The text was proofread in its entirety by an expert English-speaking scientist.

Round 3

Reviewer 3 Report

Review attached.

.

Author Response

June 6, 2023

Dears

Professor

Maeve Wang

IJMC

MPDI

A greeting and thank you for your contributions to this writing.  Professor Wang This is the third time we have reviewed our manuscript and adjusted the reviewers' suggestions in their entirety.

Apparently reviewer 3 does not receive our corrections and he cannot see the changes made, but it is the third time that he has asked us exactly the same thing, and the authors do not understand what he wants, a situation that is uncomfortable.

Response to Reviewer 3 Comments

Point 1: Please check that all references are relevant to the contents of the manuscript.

Response 1: The references have been carefully checked and it is good for the reader of this topic to keep them in this paper in case he/she wishes to expand on any aspect. Additionally, they are numbered and referenced according to IJMS standards. We will leave all the references because they support the bibliographic review carried out.

Point 2: Please provide a cover letter to explain, point by point, the details
of the revisions to the manuscript and your responses to the referees’
comments.

Response 2:

Point 3: If you found it impossible to address certain comments in the review
reports, please include an explanation in your appeal.

Response 3: Not applicable. We have solved and changed everything that has been requested.

Point 4: The revised version will be sent to the editors and reviewers.

Response 4: All requested corrections have been sent on time.

Point 5:  If one of the referees has suggested that your manuscript should undergo
extensive English revisions, please address this issue during revision. 

Response 5:  The text was proofread in its entirety by an expert English-speaking scientist.

Best Regards,

Prof. Dr. Juan Carlos Moreno-Pirajan

Full Professor

University of the Andes

Bogota, Colombia

Round 4

Reviewer 3 Report

The article can be printed in its current version.

.